# Influence of Scanning Strategy and Post-Treatment on Cracks and Mechanical Properties of Selective-Laser-Melted K438 Superalloy

Bin Zhang, Hua Yan *, Zhisheng Xia, Peilei Zhang, Haichuan Shi and Qinghua Lu

School of Materials Science and Engineering, Shanghai University of Engineering Science, Shanghai 201620, China; bin2030@126.com (B.Z.); xzsfzth@163.com (Z.X.); oxidpl@126.com (P.Z.); shc0010@126.com (H.S.); luqh@sues.edu.cn (Q.L.)
* Correspondence: yanhua@foxmail.com; Tel.: +86-216-779-1412

**Abstract:** The feasibility of manufacturing high-performance components with complex structures is limited due to cracks in some superalloys fabricated by selective laser melting (SLM). By controlling the main process parameters such as scanning strategy, the adverse effects of cracks can be effectively reduced. In this paper, the effects of two different SLM scanning strategies with island and 'back-and-forth' and post-heat treatment on the cracks and mechanical properties of selective-laser-melted (SLMed) K438 alloy were investigated. The results show that the SLM method of the 'back-and-forth' scanning strategy had better lap and interlayer rotation angles and a more uniform distribution of laser energy compared with the island scanning strategy. The residual stress accumulation was reduced and crack formation was inhibited under this scanning strategy owing to the cooling and shrinkage process. In addition, the dislocation motion was hindered by the formation of uniformly dispersed MC carbides and $\gamma'$ phases during the SLM K438 alloy process, which resulted in the density of the as-built SLMed K438 alloy being up to 99.34%, the hardness up to 9.6 Gpa, and the tensile strength up to 1309 MPa. After post-heat treatment, the fine secondary $\gamma'$ phases were precipitated and dispersed uniformly in the Ni matrix, which effectively improved the Young's modulus and tensile strength of the alloy by dispersing the stress-concentrated area.

**Keywords:** K438 nickel-based superalloy; selective laser melting (SLM); scanning strategy; mechanical properties; heat treatment; crack





## 1. Introduction

Selective laser melting (SLM) is one of the most widely used additive manufacturing (AM) technologies for metallic materials [1]. With continuous improvements in SLM technology, many alloys, including Ti alloys [2], Al alloys [3], nickel-based superalloys [4], and certain steels, can be successfully processed using SLM, leading to high densities and favorable mechanical properties [5]. The SLM process demonstrates outstanding performance in the fabrication of nickel-based superalloys and has clear advantages over other additive manufacturing processes in terms of geometric complexity, high efficiency, flexibility, and cost-effectiveness compared to conventional manufacturing methods [6–8].

With the development of SLM technology, the SLMed nickel-based alloys have been be widely used. For instance, K438 alloy components plays a vital role in both marine and aerospace industries, which must withstand high temperatures and stresses [9–11]. The K438 superalloy has complex chemical properties, including Co, Cr, Al, Ti, and Mo [12]. It is mainly composed of disordered ($\gamma$) substrates, ordered ($\gamma'$) particles, and various carbides, which are enhanced by solid solution strengthening, grain boundary strengthening, and precipitation strengthening [13,14]. Furthermore, K438 heat resistance, corrosion resistance, toughness, plasticity, and other properties and its reinforcement mechanism are closely related. The primary method of reinforcement in K438 alloys is the precipitation of

intermetallic compounds, particularly $Ni_3(Al, Ti)$, resulting in the formation of $\gamma'$ particles within the FCC phase. These alloys display remarkable properties resulting from the solid solution strengthening of Co and Mo in the matrix, enhanced oxidation resistance due to a high Cr content, the $\gamma$-phase precipitation strengthening of Al and Ti, the dispersion strengthening through MC and $M_{23}C_6$ carbides and boride-forming elements, and the combined effects of multiple strengthening mechanisms [15,16]. This combination of strengthening mechanisms gives K438 high-temperature alloys their exceptional properties. These alloys are especially favored for aero-engine blades and gas turbines due to their exceptional thermal stability and resistance to thermal corrosion [17–19]. Even at elevated temperatures ranging from 540 °C to 1000 °C, nickel-based superalloys maintain excellent mechanical and physical properties.

Although their superior resistance to high temperatures, SLMed K438 alloys are susceptible to extensive cracks caused by precipitation and cooling stresses during the rapid solidification and shrinkage stages of the forming process [20]. The high contents of Al and Ti (Al + Ti > 6%) in these high-temperature alloys, where $\gamma'$ serves as the strengthening phase, increase their vulnerability to cracking during forming, thus impacting the final material properties [21]. The primary types of cracking observed in K438 alloys are liquefaction cracking (LC), solidification cracking (SC), and ductility dip cracking (DDC) [22,23]. Liquefaction cracking mainly occurs at grain boundaries, while ductility dip cracking (DDC) results from low-melting eutectics of $\gamma+\gamma'$, carbides, and borides [24,25]. DDC complements liquefaction cracking during the growth phase. For instance, Xu et al. [26] investigated the liquefaction cracking mechanism of the high-temperature alloy IN-738LC. They observed that the semi-continuous $\gamma+\gamma'$ eutectic at the grain boundary, dispersed during the process of grain boundary liquefaction, locally melted and formed a liquefaction crack through the rupture of the liquid film. Additionally, Zhang et al. [23] explored crack propagation in the high-temperature alloy IN738 using laser melting deposition (LMD) techniques. They identified that microscopic liquefaction cracks initiated at the center along extended straight grain boundaries and propagated due to the coupling effect of LMD. In an attempt to address hot tearing at high temperatures, Heydari et al. [27] conducted a study where the element Zr (0.01 wt.%~0.04 wt.%) content in K438 high-temperature alloys was varied. The results indicated that a higher Zr content prevents grain aggregation at grain boundaries but increases the alloy's susceptibility to hot tearing. Crack failure caused by cracking is a common and very serious failure mode. Considering the influence of cracks, the methods of adjusting process parameters and post-treatment are commonly used to reduce the harm caused by cracks.

Many researchers carried out the influence of process parameters on the mechanical properties and overall quality of AM products. By varying the process parameters (laser power, scan rate, scan spacing, layer thickness, and scanning strategy), defects in the material forming process could be reduced. These defects were a critical factor in the final properties of the material [5,28]. However, few studies have changed the scanning strategy to observe the internal structural properties of alloys. As the motion pattern of the energy beam in space, the scanning strategy is coupling influence by the direction and sequence of the scanning, the vector rotation angle, and the vector length. During SLM, the high-speed motion of the energy beam leads to rapid changes in temperature distribution, which triggers high residual stresses and inhomogeneous deformation, affecting product quality [29]. Therefore, more and more studies have been conducted to enhance the product quality of SLM by optimizing the scanning strategy. Rashi et al. [30] improved the mechanical properties of 17-4PH stainless steels produced by SLM by changing the number of laser scans per layer in the scanning strategy. It was found that by using a scanning strategy of two laser scans per layer of powder, the hardness of the samples increased by 44% and the relative density was approximately 0.06%~0.08% higher. Xu et al. [31] investigated the histo-mechanical properties of 24CrNiMo low alloy steel produced by SLM using four scanning strategies and found that the continuous scanning strategy with a 90° interlayer transition tended to have an equilibrium temperature field, higher relative

density, minimal residual stress, and the best surface quality. Furthermore, Nong et al. [32] found that densification, microstructure, crystalline structure, and mechanical properties could be tailored by SLM using different scanning strategies by comparing four continuous scanning strategies with different angles on the histological properties of 15-5PH stainless steel prepared by SLM. The above researches show that scanning strategy can significantly improve the formability of SLMed superalloy, which has good scientific research value.

Moreover, post-treatment of the specimen is crucial to enhance the internal integrity of the alloy. The microstructure of an alloy significantly influences its mechanical properties. Factors such as high residual stresses, excessive internal porosity, and susceptibility to cracking can impact the final mechanical properties of the material. A suitable post-treatment program can be developed to achieve excellent mechanical properties. For instance, Cao et al. [33] examined the mechanical properties of IN718 produced by SLM and subjected to 1080 °C homogenization heat treatment. They discovered that an appropriate amount of δ phase, under the influence of the homogenization temperature, acts as a stabilizer on the grain boundaries, thereby enhancing the grain boundary strength and extending the creep rupture life of the sample. Additionally, Luo et al. [34] investigated the Inconel 718 alloy produced through selective laser melting with various heat treatments. Their findings demonstrated that a suitable heat treatment significantly improved the corrosion resistance of the specimens by reducing micro-segregation and modifying the shape of the second phase. The heat treatment of the nickel-based K438 alloy primarily involved a solid solution and aging treatment processes. The solid solution treatment dissolved all the γ′ particles in the substrate, while the aging treatment led to the reprecipitation of dense and homogeneous γ′ particles of an appropriate size. Sotov et al. [35] compared the effects of standard heat treatment and hot isostatic pressing (HIP) conditions on the behavior of the IN738 alloy. They observed an increase in the yield strength at 900 °C for both heat treatments. Wongbunyakul et al. [36] investigated the microstructural properties of the IN738 alloy with different temperatures and times of solid solution and aging treatments. They noted that decreasing aging treatment times and increasing solid solution treatment temperatures resulted in a consistent decrease in the size and area fraction of γ′ particles, resulting in finer γ′ particles and hardness values above 450 HV.

Thus, this study focuses on the effects of different scanning strategies and post-treatment on the density of existing K438 alloy materials, type of crack production, characterization of the microstructure, and mechanical properties. In particular, the relationship between the two scanning strategies and crack formation in the printing process was systematically studied. The microstructure was used to deeply analyze the root causes of cracks and the internal mechanism leading to differences in mechanical properties. On this basis, the subsequent effects on cracks and mechanical properties were further explored through post-treatment. The findings of this research can contribute to the advancement of selective laser melting technology by providing guidelines for optimizing the processing parameters and post-treatment methods to achieve the superior short columnar dendritic microstructure and higher hardness and tensile strength of K438 alloy samples. Ultimately, this can lead to the development of high-performance products with improved reliability and durability, benefiting industries that rely on K438 alloys for their applications.

## 2. Experiments

### 2.1. Fabrication of K438 Alloy

The K438 alloy powder was prepared using gas atomization and supplied by Avimetalam. Table 1 provides the chemical composition and composition content of the K438 alloy powder. The particle size distribution of the powders was determined by laser diffraction: 19.0 μm, 33.3μm, and 56.2 μm for D10, D50, and D90, respectively. It was observed by SEM that most of the powders had good sphericity and a smooth surface, and a few powders had depressions or protrusions, but this did not affect the overall fluidity of the powder, as shown in Figure 1.

**Table 1.** Chemical composition and content of K438 alloy.

| Element | C | Ta | Mo | W | Ti | Al | Co | Cr | Ni |
|---|---|---|---|---|---|---|---|---|---|
| Content (wt.%) | 0.1–0.2 | 1.5–2.0 | 1.5–2.0 | 2.4–2.8 | 3.0–3.5 | 3.2–3.7 | 8.0–9.0 | 15.5–16.3 | Bal. |

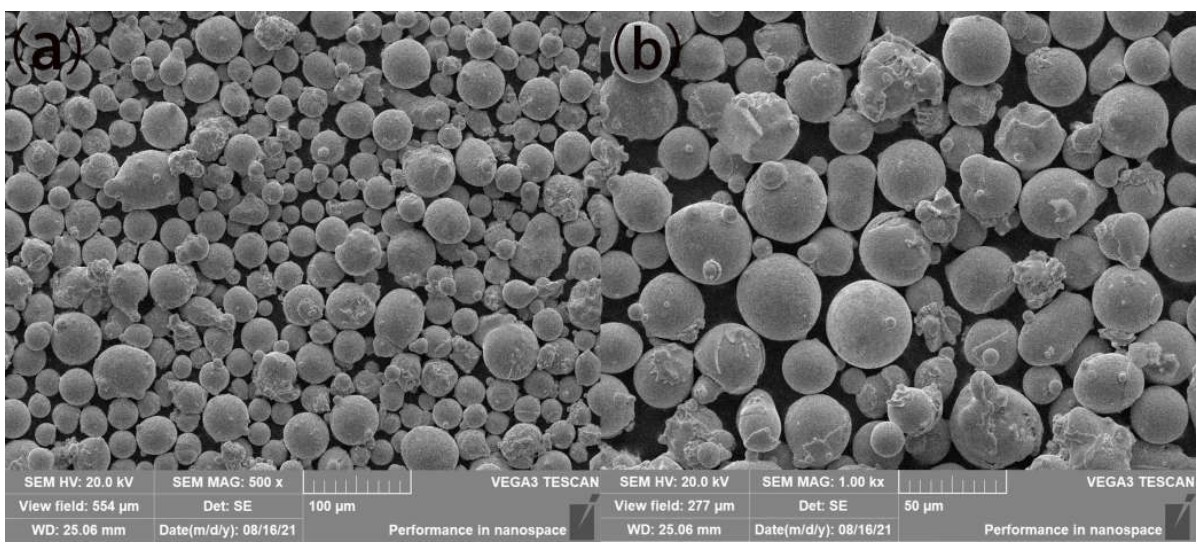

**Figure 1.** SEM of K438 powder at different magnifications, (**a**) low power SEM and (**b**) enlargement SEM.

The SLMed K438 alloy specimens were printed by a machine named M2 cusing system (Concept Laser Company, Bayern, Germany). The laser was an IPG fibre laser with a maximum power of 400 W and a spot diameter of 270 μm. Scanning processes using a high-speed scanning scope controlled forming accuracy. An isolated air precision powder feeding structure controlled the oxygen content below 50 ppm using highly pure nitrogen (99.99%) for protection. The SLM equipment schematic diagram is shown in Figure 2.

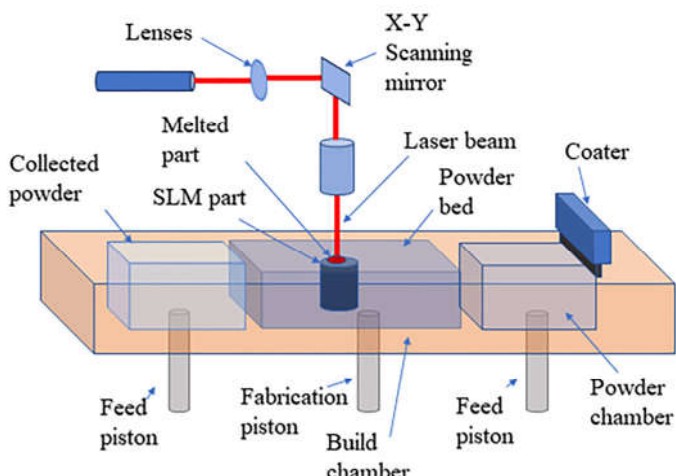

**Figure 2.** Equipment schematic of SLM.

*2.2. Experimental Parameters*

In this experiment, the following laser parameters were used: laser power of 370 W, scanning speed of 700 mm/s, scanning spacing of 0.13 mm, and spot diameter of 270 μm. The SLMed K438 alloy sample is shown in Figure 2a. As shown in Figure 3c, two different scanning strategies were used to SLM with the same basic process parameters. The specimens with the 'back-and-forth' scanning strategy were labeled as S1 while those with

the island scanning strategy were labeled as S2. 'Back-and-forth' scanning strategy is 67° rotation of the adjacent layer scanning vector and all scan lines parallel to each other. This scanning strategy implemented a non-regional, evenly distributed diagonal scanning method. Island scanning strategy is 90° rotation of scan vector between adjacent islands. After completing one region, the next region was scanned along the diagonal direction to reduce the laser energy concentration. This process was repeated until one layer of printing was completed. This scanning strategy divided the printing area into rectangular areas evenly, and the scanning directions of adjacent areas were perpendicular to each other.

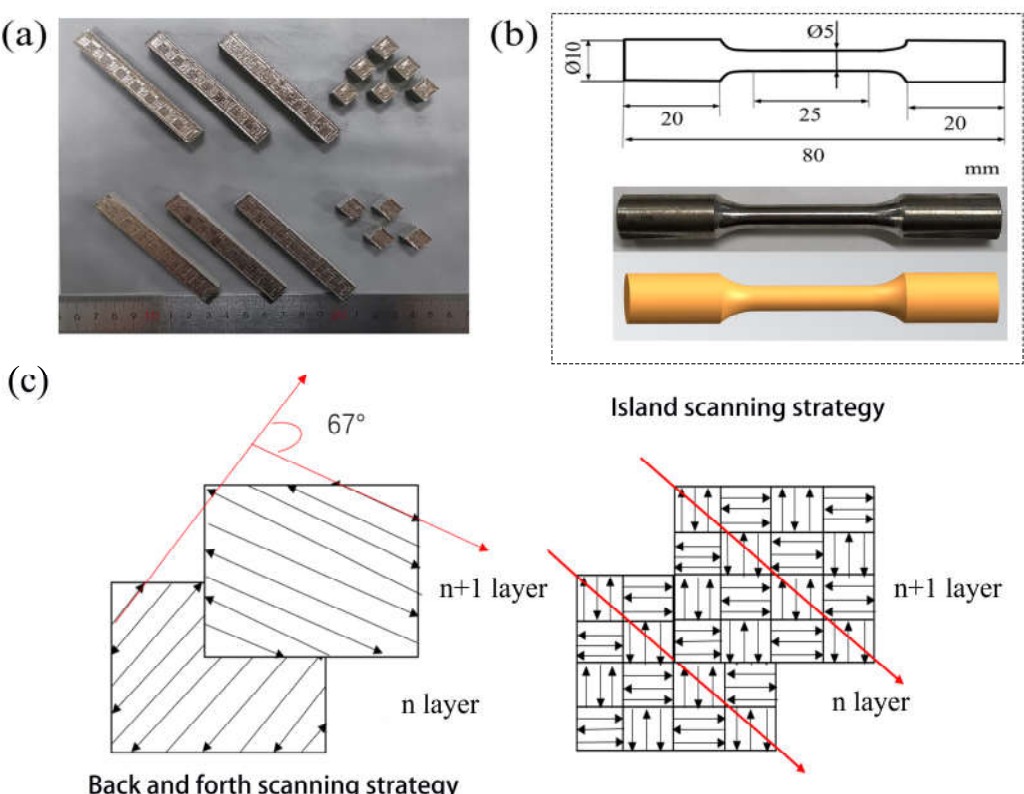

**Figure 3.** Schematic diagram of scanning strategies: (**a**) sample preparation of as-built samples; (**b**) tensile test sample size; (**c**) 'back-and-forth' and island scanning strategies.

The post-treatment procedure consisted of two steps. First, the specimens were subjected to solid solution treatment at 1120 °C for a duration of 2 h and subsequent air cooling. The aging treatment was carried out at 850 °C for a duration of 24 h and air-cooling subsequently. The post-treated S1 and S2 specimens are denoted by HTS1 and HTS2, respectively. The standard process parameters of K438 were used for the standard post-treatment (solid solution treatment and aging treatment) of the samples after employing different scanning strategies to observe the tissue evolution and mechanical properties after the different scanning strategies and post-treatment.

In addition, oxide films and impurities were removed from the substrate surface using a sandblaster on a substrate made of nickel-based alloy with a thickness of 20 mm. The substrate was kept at a temperature of 200 °C during the whole preparation and manufacturing process. At least two exposures were made in order to minimize heat dissipation gradients and prevent warping, chipping, and distortion when melting the initial layer of powder. Two different scanning strategies (island scanning strategy and 'back-and-forth' scanning strategy) were used to prepare a square sample to study the microstructure of a K438 alloy with a size of 10 mm × 10 mm × 10 mm. In addition, in order to study its tensile properties, a rectangular block with a size of 10 mm × 10 mm × 80 mm was prepared and processed into a round rod-shaped tensile specimen that met the requirements,

as shown in Figure 3a,b. The SLMed samples were separated from the substrate plate by wire electrical discharge machining. After post-treatment, mechanical lapping was used for preparing microstructure observation samples and then mechanically polished with a 50 nm particle-sized suspension. The samples were then etched in a Kalling etchant (11 m $C_2H_5OH$, 11 mL HCl acid, 0.5 ml $CuCl_2$) for 30 s to observe the morphology, followed by immersion in an etchant (25 g $FeCl_3$, 25 mL HCl, 100 mL $H_2O$) for 20 s to observe the phase morphology after etching. Subsequently, the sample was cleaned with alcohol using an ultrasonic cleaner for 30 min to remove surface impurities. After drying, the samples were placed in a drying oven for tissue observation.

### 2.3. Characterization

The experimental samples were analyzed to determine their density, hardness, tensile properties, and microstructural characteristics using various scanning strategies and post-treatment. The density of the K438 alloy was determined using Archimedes' method by the average of the five measurements with a balance precision of $\pm 0.0001$ g. Microstructure analysis was conducted using the optical microscope, scanning electron microscope (SEM) equipped with energy dispersive spectrometry (EDS), and electron backscatter diffraction (EBSD). X'pert Pro X-ray diffraction (XRD) was employed to examine and analyze the phase composition of the samples, using a Cu target with a goniometer movement step of $0.02°$ and a test angle range of $10°$ to $100°$. Additionally, continuous stiffness measurements (CSMs) were performed using the nanoindenter, employing a standard Brinell diamond indenter probe. The measurements were carried out using a constant strain rate of 0.05/s. The experiments involved an approximate displacement of 800 nm (using a 100 μm radius indenter), a maximum load of 50 mN, and a dwell time of 20 s. To ensure consistency in variables, five tensile specimens were tested with all specimens oriented transversely parallel to the substrate.

## 3. Results and Discussion

### 3.1. Density

Figure 4 illustrates the trend in alloy density under various scanning strategies and post-treatment conditions. The densities of the alloys were calculated by equation $\rho = \rho_p / \rho_0 \times 100\%$ for different scanning strategies, with a theoretical density of 8177.1 $kg/m^3$. $\rho_p$ is the measured density of the specimen block and $\rho_0$ is the theoretical density of the K438 alloy. Significant differences in density between the scanning strategies were observed with a difference of 0.21%, and the 'back-and-forth' scanning strategy (S1) resulted in a higher density of 99.34%, as opposed to 99.13% for the island scanning strategy (S2).

The use of the 'back-and-forth' scanning strategy resulted in higher densities. This can be attributed to the unique role of the 'back-and-forth' scanning strategy in the process of scanning. Rotating the direction of the scanning vector by $67°$ between adjacent layers resulted in a more uniform heat flow rotation and the alternate formation of a molten pool, leading to a uniform temperature gradient and complete melting of the particles, resulting in higher densities [37,38]. The island scanning strategy often started or ended the laser run at the overlap. The acceleration or deceleration of the laser at these locations resulted in it staying for longer, causing an increased local heat input at this point [39]. This led to the formation of a deeper molten pool, making it difficult for gas to escape from the melt. As a result, the overlap may have contained more small pores in comparison to other regions of the formed part. The existence of these pores eventually reduced overall density.

As shown in Figure 5, in laser scanning, due to the Gaussian energy distribution of the laser beam, the central energy is higher and the peripheral energy is lower. The solidified single-track molten pool also has a higher center and lower edges. In contrast to the island scanning remelting mode (no inter-layer rotation), the use of the 'back-and-forth' scanning strategy with a $67°$ rotation between each layer compensated for the height difference between the center and the edges caused by scanning the previous layer. This

ensured that the overall height difference across the sample surface remained less than the thickness of the single layer of powder, resulting in a more uniform powder distribution and higher density. Inter-layer scanning modes with different scanning strategies caused different melting and solidification modes within the alloy, resulting in different melting and residual stress distributions due to the rapid melting and solidification processes of the alloy. Such differences within the alloy also led to different formations of defects such as cracks, pores, and unmelted pores. Therefore, in the experiments, different scanning strategies led to differences in the densities of the alloys, even if other process parameters were kept constant.

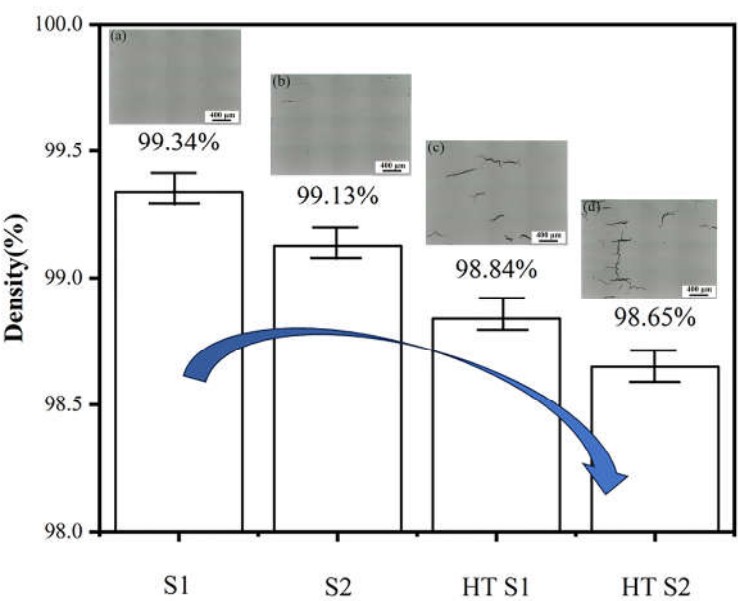

**Figure 4.** Density of different scanning strategies and post-treatment, (**a**) Surface OM of S1, (**b**) Surface OM of S2, (**c**) cracks of HT S1 and (**d**) cracks of HT S2.

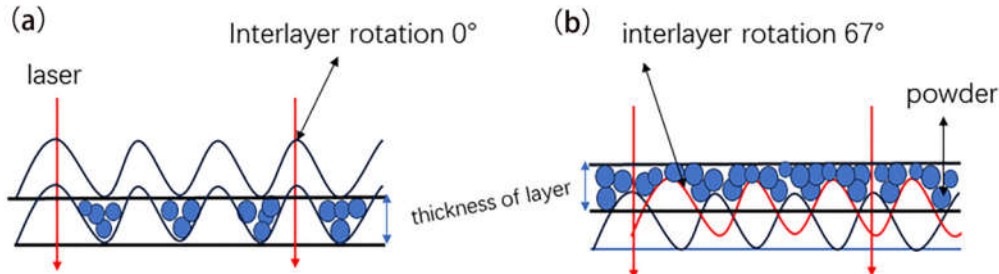

**Figure 5.** Schematic diagram of powder with different interlayer rotation angles under the same layer thickness, (**a**) island scanning remelting mode and (**b**) 'back-and-forth' scanning mode.

As a result of the post-treatment, the density of the alloys with the island scanning strategy and the 'back-and-forth' scanning strategy was reduced by 0.5% and 0.48%, respectively. During the post-treatment process, the crack morphology changed from fine to coarse, and the number of cracks increased significantly. After the aging treatment, recrystallized grains were generated near the original grain boundaries and inside the grains. As the size of the grains and grain boundaries increased, cracks were generated at the grain boundaries and propagated along the grain boundaries. At the same time, the solid solution treatment released residual stresses, contributing to the formation of microcracks. These microcracks could extend and penetrate the interior of the material to form pores and the cracks could expand into larger coarse cracks, leading to a decrease in the density of the post-treatment alloy.

### 3.2. Distribution and Types of Cracks

In general, by varying fundamental process parameters such as laser power and scan speed, one can regulate the depth and width of the molten pool, influence its morphology, adjust the melting efficiency, and influence the size of the solidification grains. These parameter variations directly affect the microstructure and properties of the material, reducing internal alloy defects while maintaining consistent formation patterns. Consequently, by independently modifying the scanning strategy of the alloy and subsequently modifying the laser scanning path and the overlap pattern, it becomes possible to more effectively discern the influence of different scanning strategies on internal crack defects within the alloy at the microscopic level. To further investigate the effects of different scanning strategies leading to internal cracks in the alloy, SEM images showing crack trends after different scanning strategies and post-treatment were analyzed as shown in Figure 6.

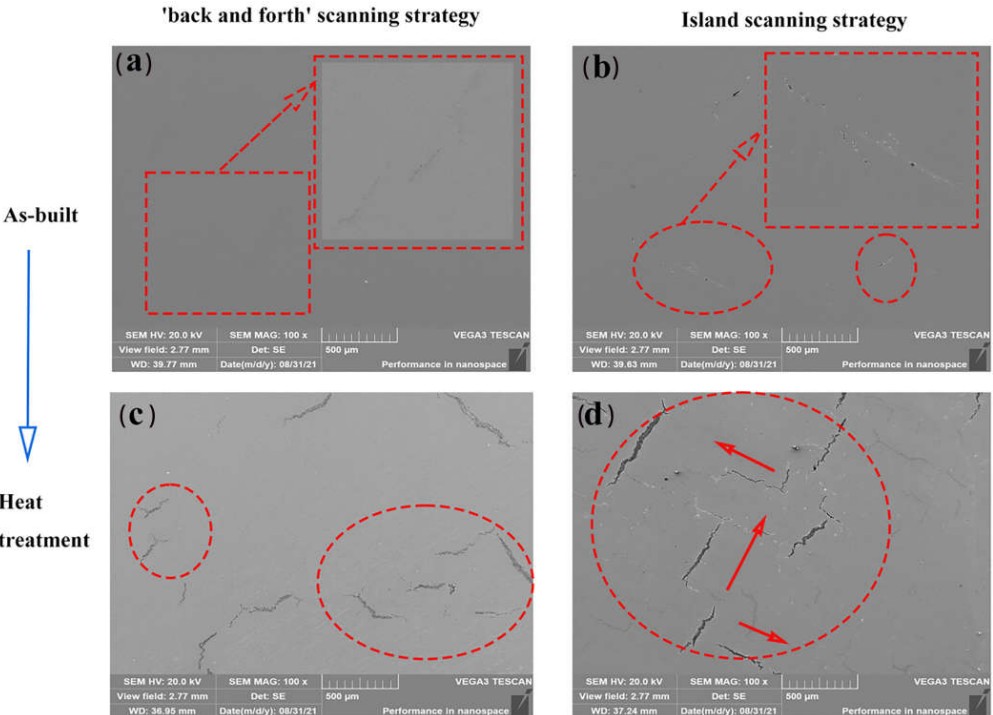

**Figure 6.** SEM images of cracks trend after different scanning strategies and post-treatment, (**a**) cracks of S1, (**b**) cracks of S2, (**c**) cracks of HT S1 and (**d**) cracks of HT S2.

The alloys exhibited good forming quality when utilizing the 'back-and-forth' scanning strategy. The resulting surfaces exhibited smooth and uniform characteristics. The surface of the alloy revealed no visible holes or defects. The present micro-cracks, uniformly oriented, had sizes ranging from 182 to 245 μm, as shown in Figure 6a. This is because the laser vector of the 'back-and-forth' scanning strategy was parallel and opposite in many directions to maintain the residual stress at a low level, the distribution tended to be average, and the cracks were not easy to produce [40]. In contrast, utilizing the island scanning strategy increased the number of defects within the K438 alloy, as well as the appearance of pores and keyhole defects. Due to the remelting of the substrate in the overlap during the laser scanning process and concentration effect of the energy in the lap region, irregularly shaped and sized keyhole defects were generated and remained in the next layer. Two different types of vertically extending cracks were observed, with perpendicular directions, both approximately 250 μm in size, as illustrated in Figure 6b.

A comparison of the crack characteristics with the two scanning strategies reveals significant differences in their crack extension directions. The crack extension direction within the alloy had a close relationship with the laser scanning strategy, instead of being

randomly generated. The explanation for this phenomenon has two aspects. Firstly, in the 'back-and-forth' scanning strategy, the scanning directions within the same layer were parallel and opposite to each other. In this strategy, after completing the previous scanning vector, the next scanning vector was initiated in the higher-temperature region. This led to decreasing the temperature gradient and reducing the residual stress [41]. However, as the scanning process proceeded towards the central region, the laser single-scan path became longer, leading to an increase in the temperature gradient and a gradual increase in the residual stress [42]. This produced a number of cracks in the same direction. The multi-partition scanning mechanism in the island scanning strategy resulted in a shorter single laser scanning path compared to the non-partitioned 'back-and-forth' scanning strategy. This led to a more even melting and solidification rate of the powder at different locations, resulting in lower residual stress. Furthermore, repetitive melting and solidification caused a high laser energy input, which increased the thermal contraction effect of the liquid phase in the molten pool, further intensifying the level of residual stress [43,44]. While the temperature gradient inside individual islands was low, it became high at the boundaries where cracks were more likely to form in the lap region. In the island scanning strategy, the laser scanning paths were perpendicular to each other at the lap region, which aligned with the direction of the crack vector. Hence, the length of scanning vectors and the lap pattern have a direct impact on crack formation in this alloy, and there exists a strong relationship between the scanning vector and the crack vector [45].

The surface crack states obtained after post-treatment with two different scanning strategies are shown in Figure 6c,d. The number of cracks on the alloy surface increased after the post-treatment, and originally elongated cracks gradually developed into coarser and longer structures, as observed. Fine, irregularly orientated short cracks with a size of about 200–500 μm also surrounded these coarse cracks. This phenomenon can be attributed to the effect of the solid solution aging treatment, which resulted in the melting of the original coarse grains in the alloy and the precipitation of fine recrystallized grains in the grain boundary regions. The presence of recrystallized grains increased the number of grain boundaries and, when internal stresses were applied to the material, these stresses tended to be concentrated at the grain boundaries. Since grain boundaries were regions of discontinuity in the crystal structure, this stress concentration led to plastic deformation near the grain boundary and stress concentration cracks.

Cracking is a common defect problem during selective laser melting under conditions of high temperature, high stress, welding, and solidification. The main cracking types include liquefaction cracking (LC), solidification cracking (SC), ductility dip cracking (DDC), and strain-age cracking after post-treatment. Liquefaction cracking is a common type in high-temperature alloys, and its occurrence is mainly related to the segregation of the alloy composition. Using the K438 alloy as an example, during fast cooling, elements that form $\gamma'$ phases, such as Al and Ti, accumulate in the intergranular liquid phase and precipitate $\gamma'$ phases and $\gamma$-$\gamma'$ eutectic mixtures at grain boundaries, resulting in localized variation in composition. This deviation in composition results in a non-uniform distribution of composition in the solidification area, and the inhomogeneous $\gamma'$ phase tends to diffuse into the matrix and decompose again during melting. Due to the high cooling rate, the diffusion effect is limited. Therefore, MC or $\gamma$ phases that would normally be present are now surrounded by a high concentration of MC or $\gamma$ phases, resulting in the formation of a thin film consisting of a low melting point eutectic composition at the grain boundaries. As shown in Figure 7, the formation of the above-mentioned film contributes to the liquefaction of the grain boundaries during melting. However, the liquefied grain boundaries are unable to withstand the stresses, leading to the formation of liquefaction cracks [46].

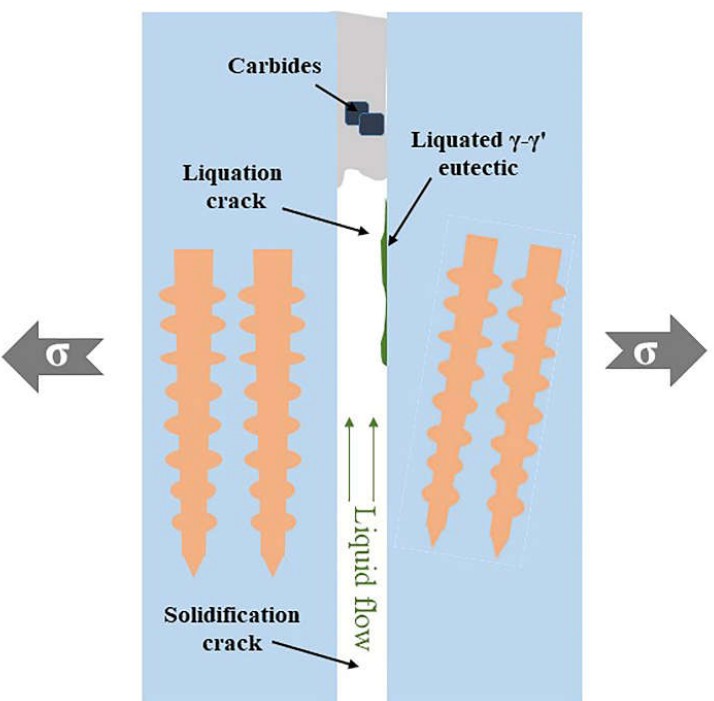

**Figure 7.** Schematic diagram of the initiation of liquation and solidification cracks in the overlapping zone.

The initiation stage of liquation and solidification cracks in the overlapping zone was examined. Solidification cracks are formed during the cooling and solidification of the liquid molten pool. They are distributed between the grain boundaries of the solidified grains or between the cellular crystals and extend along the coarse cellular crystal tissue to form long cracks. Solidification cracks are related to the residual stress state, solidification, and segregation of the alloy. In Figure 8a, the residual stresses at the laser lap are high due to the internal laser lap of the alloy. The greater the solidification temperature range of the alloy, the more likely it is that solidification cracks will form. Figure 8a illustrates that solidification cracks are closely linked to the residual stress, solidification processes, and segregation phenomena within the alloy. Solidification cracks in nickel-based, high-temperature alloys mainly stem from inter-dendrites of high-melting-point carbides and shrinkage holes formed in the final stage of solidification. The overlap of the laser in the alloy results in relatively high residual stress in the region of overlap. It is worth noting that the formation of solidification cracks is more probable in alloys with a wider solidification temperature range. During the cooling and solidification of the liquid molten pool, these solidification cracks appear and are distributed along the grain boundaries of the solidified grains or between cellular crystals, eventually propagating along the coarse cellular crystals to form long, extended cracks.

Figure 8b,d demonstrate the common characteristics of ductility dip cracking. This phenomenon typically takes place within the heat-affected area of reheated weld metal or base metal at homologation temperatures ranging from 0.4 to 0.9 [47]. These kinds of cracks are identified by edges that mostly feature straight and long triangular grain boundaries, as well as serrated grain boundaries and slip cracks adjacent to carbides or eutectics. In metals, ductile tilt cracks usually stem from grain boundaries formed by liquefaction cracks that emerge when alloy grain boundaries experience considerable residual stress exceeding the bond strength of the interface. Ductility dip cracks are usually short and thin-shaped. The formation of such cracks is due to multiple factors, including the macroscopic thermal stresses that arise during the welding process, solidification stresses, and local stresses generated by the precipitation of $M_{23}C_6$ carbides at grain boundaries [48].

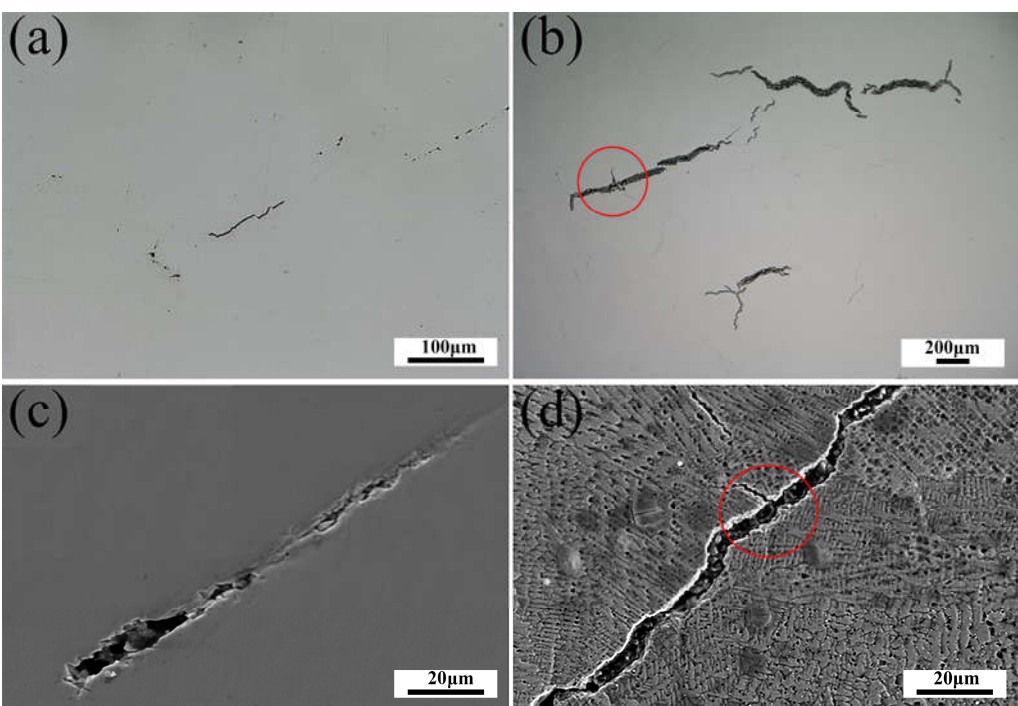

**Figure 8.** Cracks by OM and SEM: (**a**,**c**) S1; (**b**,**d**) HT S1.

*3.3. Microstructure Characterization*

Significant changes in microstructure were observed after changing the scanning strategy for the selective laser melting of the K438 alloy. In Figure 9a,a', it can be seen that the 'back-and-forth' scanning strategy, with the laser scanning perpendicular to the direction of the build, resulted in a specific microstructure with parallel orbital structures. Within the single-track structure, we could observe the presence of perpendicularly oriented cellular crystals near the edges of the tracks and a homogeneous cellular crystal structure in the center of the tracks. Due to the adjacent layer rotation of 67° in the 'back-and-forth' scanning strategy, the size of different columnar dendrites changed with the scanning direction. The elongated cellular crystals were remelted by the laser in subsequent layers, resulting in shorter columnar dendritic structures with lengths of 5–15 μm. This columnar dendritic structure tended to have more grain boundaries and finer grains, effectively inhibiting crack propagation. Furthermore, the grain boundaries, as regions of lower strength in the material, could play a role in inhibiting crack propagation. Fine grain structures could also limit the movement of dislocations at the grain boundaries, further improving the crack propagation resistance of the material. When adjacent tracks were scanned by the laser, local heating led to thermal expansion and cooling led to the contraction of the material. When the heat-affected regions of two adjacent tracks overlapped, compressive stresses caused by thermal stress and cooling contraction developed between the adjacent regions. This compressive stress helped to tightly bond the material around the grain boundaries, reducing the propagation of grain boundary defects and cracks.

As shown in Figure 9b,b', we could clearly observe distinct melting channel boundaries that were perpendicular to each other. This was typically characterize an island scanning strategy. The island scanning strategy divided the scan region into multiple islands, and each island was a radiation unit with different radiation directions [44,49]. As a result, the island scanning strategy dissipated heat faster than a conventional unidirectional scanning strategy, and heat did not accumulate significantly between regions, so the quality of the forming inside each island was better. However, in Figure 9b, some small cracks through the grain perpendicular to the grain boundary is revealed by SEM. The formation of these cracks was related to the presence of a 90° transition in the island scanning strategy, which

led to an increase in the temperature gradient in the lap region and a consequent increase in residual stress, thus contributing to crack formation.

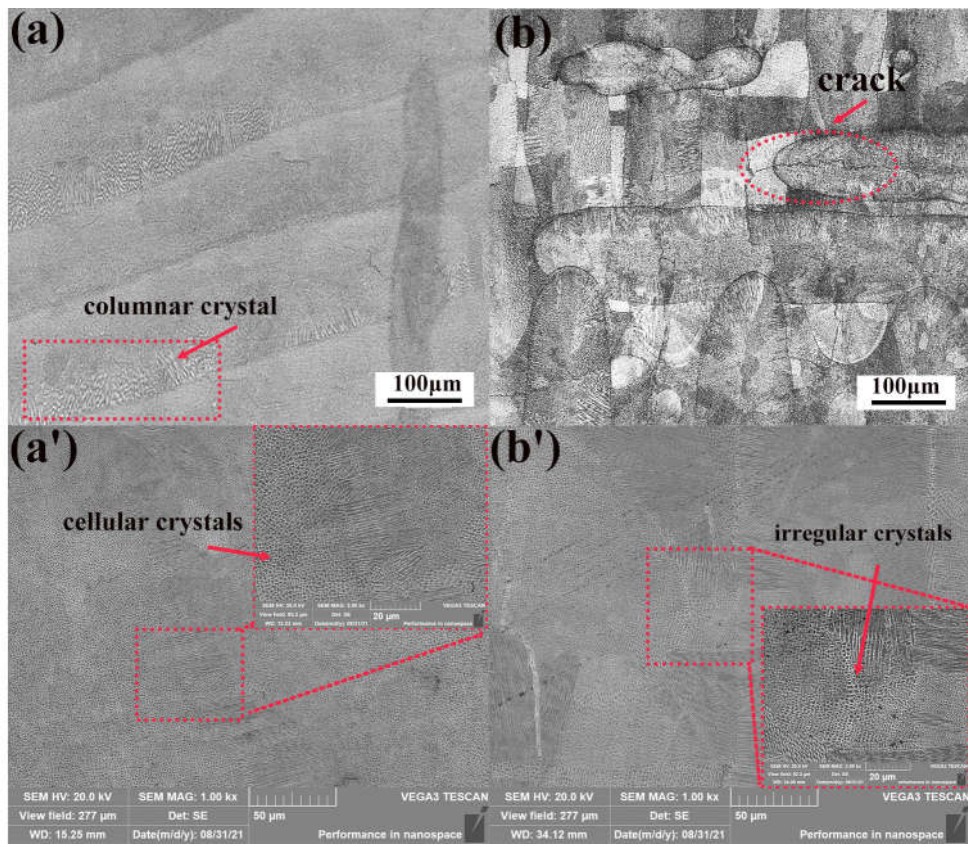

**Figure 9.** Metallographic structure and SEM images of samples under different scanning strategies: (**a**,**a'**) S1; (**b**,**b'**) S2.

When low melting point eutectic phases are present in the grain boundaries of high-temperature, nickel-based alloys, carbide precipitation can occur during cooling due to their high thermal expansion coefficient. As a result, large shrinkage stresses are generated during melting and solidification, and these stresses can initiate cracking along the solid–liquid interface and the formation of small solidification cracks near the grain boundaries [50]. In this case, the two directions of the melting channels are connected, but the connected regions show a mixed morphology of columnar and cellular crystals. Due to the characteristics of the island scanning strategy, where each island undergoes rapid solidification through different cooling directions, a mixture of short and thin cellular crystals as well as cellular crystals with different orientations and irregular morphology are produced.

Figure 10 show the micrographs under both scanning strategies after standard post-treatment. As can be seen from the Figure 10, the orbital structure of the two scanning strategies were disappeared and the grain growth was affected by the migration of the grain boundary after post heat treatment processing. However, a large number of refined recrystallized crystals (columnar subcrystals, honeycomb subcrystals) emerged near the original crystal boundary. The reason for this phenomenon is due to the decrease in the grain boundary resulting the binding force could not support the internal stress and the dislocations moved to the grain boundaries. A large number of dislocations causes the orientation difference of the grain boundary to increase and the proportion of the apparent angular crystal boundary ($\theta > 15$) gradually increase. Those refined recrystallized crystals hindered the growth of recrystallized grains and causing irregular forces at the grain boundary.

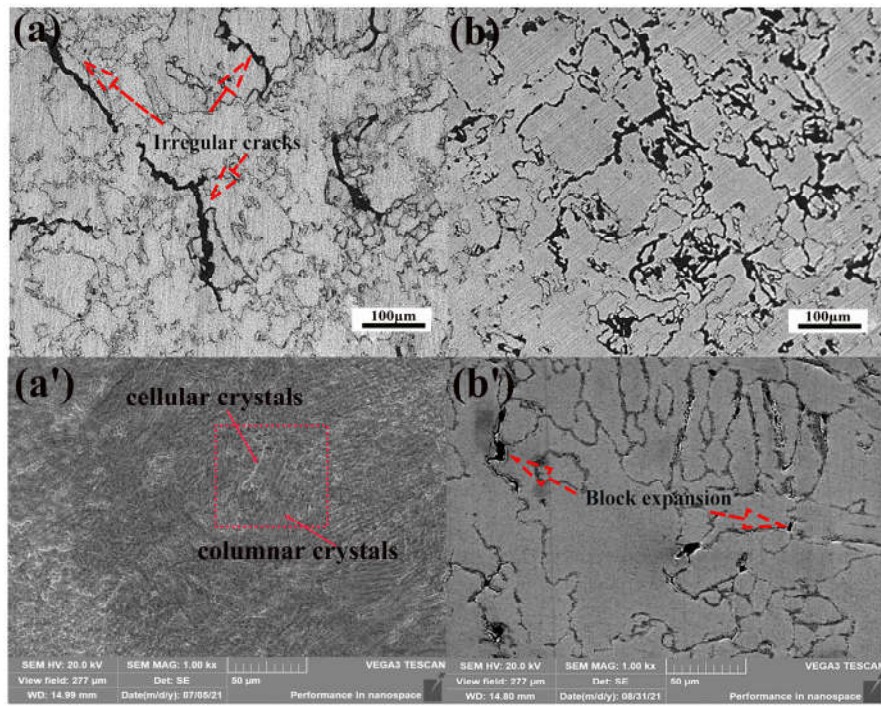

**Figure 10.** Metallographic structure and SEM images of samples subjected to various post-treatment conditions: (**a**,**a′**) HT S1; (**b**,**b′**) HT S2.

After aging treatment, elements such as Al, Ti, and Ni in the alloy precipitated from the grain boundaries and formed a large number of $M_{23}C_6$ carbides with the carbon in the matrix, forming banded regions of carbide distribution. This carbide precipitation adversely affected the plastic deformation capability of the grain boundary region, making it more brittle and reducing the plasticity of the grain boundary. When external stress was applied to the grain boundary region, cracks easily formed and propagated along the grain boundary due to the difficulty of plastic deformation in the brittle grain boundary. In addition, phase transformations during heat treatment were another important factor in crack formation, such as the dissolution of the γ′ phase during solid solution treatment and the precipitation of the secondary γ′ phase during aging treatment. These phase transformations led to changes in grain structure and size, causing an accumulation of stresses. If these stresses exceed the strength limit of the alloy, cracks can form from the phase interface. However, residual stresses in the alloy can also be released by cracks. Once a crack has formed, the thermal expansion of the alloy is limited and energy is rapidly released through the crack expansion region, reducing the temperature gradient and thus slowing the rate of crack expansion. This inherent stress release can counteract to some extent the crack expansion caused by stress build-up, helping to limit further crack expansion.

Figure 11 displays the orientation image microscopy (OIM) maps in the XY plane, based on the inverse pole figure (IPF), for both scanning strategies and post-treatment [51]. In this representation, the [001] orientation is depicted in red, the (101) orientation in green, and the (011) orientation in blue. The grain structure of both scanning strategies primarily consisted of columnar and equiaxed crystals in the as-built state. As illustrated in Figure 11a, under the 'back-and-forth' scanning strategy, the cellular crystals exhibited a pronounced (001) orientation. Notably, different particles exhibited different color changes, indicating that the rotation angle between continuous layers led to significant orientation differences. On the other hand, in the island scanning strategy depicted in Figure 11c, attributed to the substantial 90° orientation difference between the scanning directions of adjacent islands, coupled with directional solidification within each island, the grains exhibit different orientations due to the variation in cooling rates among neighboring

cellular crystals. Additionally, at the grain boundaries of the cellular crystals, there was the presence of unevenly sized equiaxed crystals. Following the standard post-treatment, the number of cellular crystals noticeably increased, and a considerable population of recrystallized grains with comparable colors emerged within the sample.

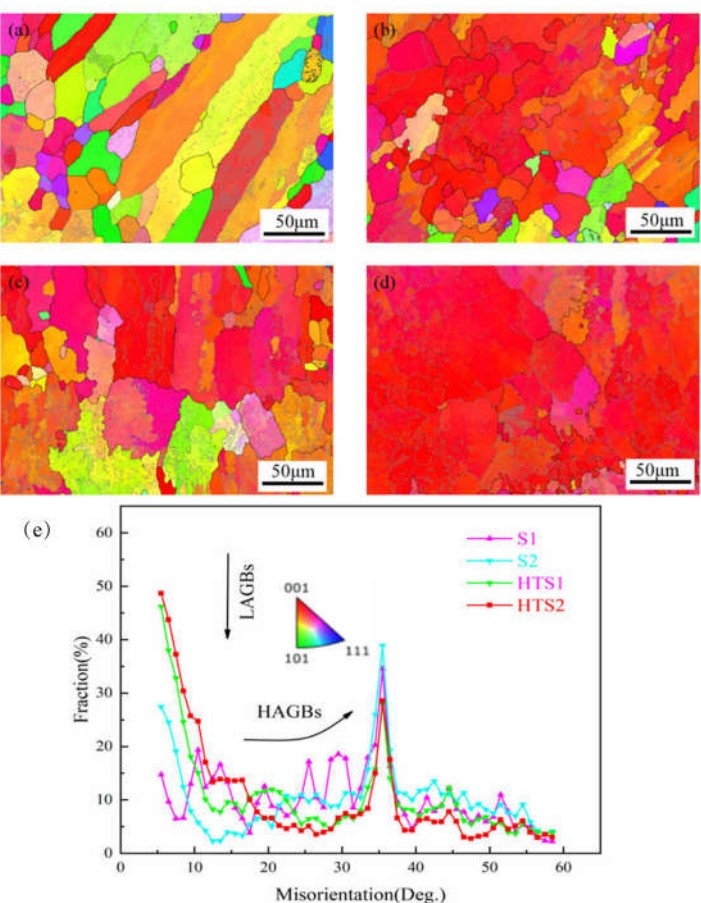

**Figure 11.** OIM images in the X-Y plane based on different scanning strategies and post-treatment conditions: (**a**) S1; (**b**) HT S1; (**c**) S2; (**d**) HT S2; (**e**) misorientation angle distribution.

Due to minimal grain deformation, the recrystallized grains exhibited a low orientation gradient. While the coarse grains underwent SLM and aging treatment, resulting in the formation of uniformly sized equiaxed grains, as illustrated in Figure 11b,d. Grain boundaries displaying misorientation angles ranging from 2° to 15° were classified as low-angle grain boundaries (LAGBs), whereas those with larger misorientation angles were considered high-angle grain boundaries (HAGBs). Dislocations migrated along the grain boundaries, being absorbed by both the grain boundaries and the LAGBs, thus leading to the conversion of LAGBs into HAGBs [52]. Notably, the significant peak increase in HAGBs occurred at approximately 35°. The HAGBs exhibited numerous intricate agglomerates, and fine equiaxed grains were observed within the grains. Subgrain nucleation were mainly occurred in the subgrains, as shown in Figure 11e. After post-treatment, the smaller misorientation meant that the orientation difference of grain boundaries was smaller and the grain boundaries were more stable. In addition, the size of the misorientation also affected the behavior of crystal defects and grain boundary migration in the crystal. The smaller misorientation could reduce the dislocation density of the grain boundary, thereby improving the tensile strength of the material.

K438 is a precipitation-strengthened alloy, with its strengthening effect mainly achieved through the dislocation hindering mechanism of $\gamma'$ phases and carbides. To verify the presence and variation of the strengthening phases and carbides in the alloy, both the

as-printed alloy and heat-treated samples were analyzed using X-ray diffraction (XRD) patterns, as depicted in Figure 12c. Upon comparison with the physical phase cards in the standard physical phase database (PDF database), several well-defined diffraction peaks were detected in the prepared samples. These peaks corresponded to the matrix γ phase of the FCC structure and the (111), (200), and (220) crystallographic planes of the reinforced γ′ phase of the K438 alloy. SEM images were used to observe the morphological evolution of the γ′ phase under the 'back-and-forth' scanning strategy. During the as-built process, the austenitic matrix underwent the transient nucleation and growth of γ′ phases due to rapid melting and solidification induced by SLM and higher laser temperature. The γ′ phases in the alloy exhibited a square structure. Variations in cooling rates resulting from different scanning strategies caused differences in the sizes of γ′ phases in different parts of the alloy. The larger γ′ phase, illustrated in Figure 12a, measured approximately 8.5 μm and was surrounded by several nanoscale, fine γ′ phases.

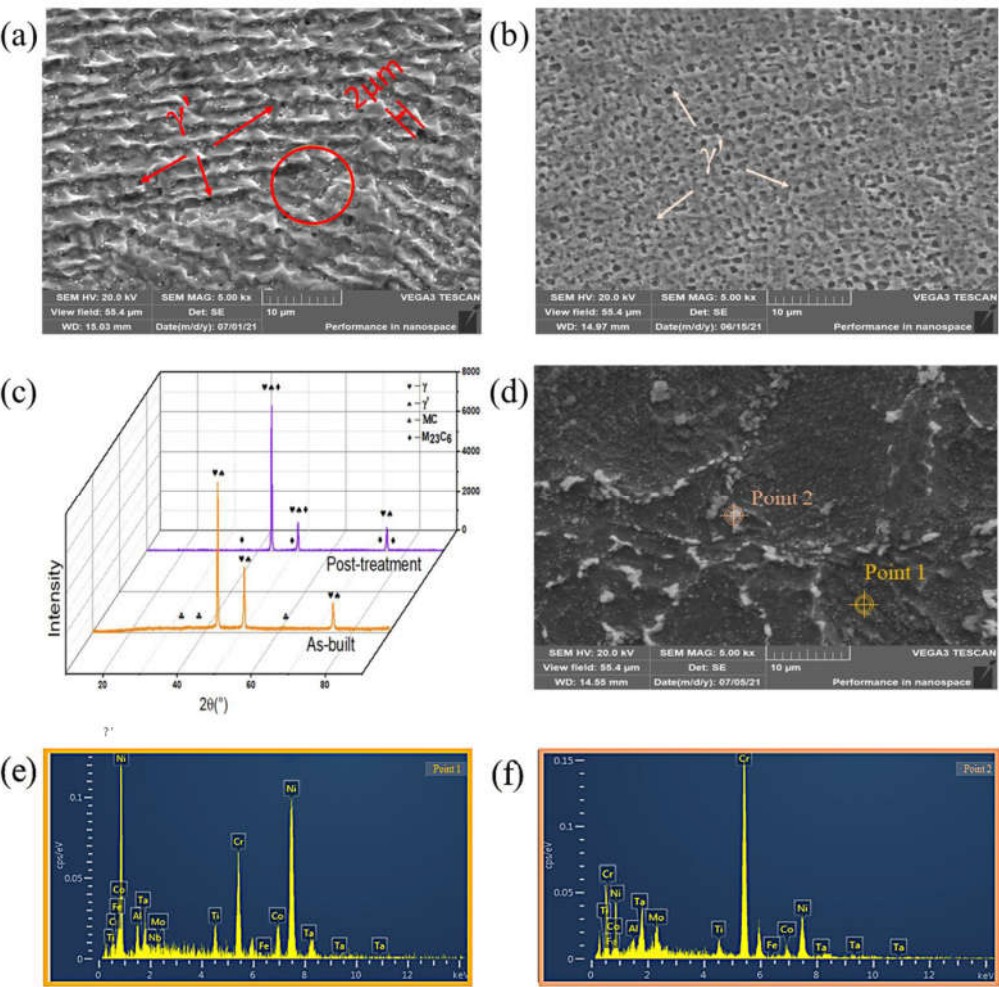

**Figure 12.** 'Back-and-forth' scanning strategy γ′ phase evolution: (**a**) print state; (**b**) post-processing; (**c**) XRD diffraction pattern of the specimen post-treatment; (**d**–**f**) EDS analysis of carbide of HT S1.

After post-treatment, the γ′ precipitated phases melted at around 1170 °C. The coarse square γ′ phase within the material dissolved, resulting in the precipitation of a homogeneous mixture of fine γ′ phases, consisting of both square and round shapes, with an average size of approximately 0.1 μm, as shown in Figure 12b. The interfacial and strain energies influenced the variation of the shape of the γ′ phase in the solid solution. At low temperatures, there was an increased mismatch between the γ and γ′ phases, and the shape of the γ′ phase was primarily influenced by the strain energy. Since the strain energy was

smaller for a square compared to a sphere, the γ′ phase exhibited a regular square shape. Conversely, at high temperatures, the mismatch between the γ and γ′ phases decreased, and the shape of the γ′ phase was predominantly determined by the interfacial energy. As the interfacial energy of a sphere is lower than that of a square, the γ′ phase tended to adopt a spherical shape.

The generation and distribution of carbides significantly affected the overall performance of the K438 alloy during its selective laser melting and post-treatment process. For this reason, it was crucial to conduct an in-depth study of how carbides were generated and distributed. The K438 alloy commonly had two types of incipient carbides that were present at the grain boundaries and within its grains during initial processing and post-treatment. The presence of such carbides had a dual effect: the pinning effect of dislocations hindered the movement of dislocations, strengthening the alloy. Additionally, due to the accumulation of carbides in the grain boundary region, the plasticity of the grain boundaries was reduced, internal stresses within the alloy increased, and grain boundaries became prone to cracking. After the post-treatment of the alloy, continuous white carbides were found to aggregate along the grain boundaries. EDS analysis reveals that the continuous carbides and the fine granular carbides are dispersed in the grains of HT S1, as shown in Figure 12d–f. Combined with XRD analysis shows the presence of Ta, Mo, Nb, and Ti elements, which readily combined with C elements to form MC carbides after post-treatment, as shown in Figure 12c.

Furthermore, the EDS of the diffuse white particles in Figure 12d shows that these elements, which are prone to MC carbide formation, were mainly distributed in the white particles, indicating that MC carbides were diffusely distributed in the matrix and could effectively inhibit cracking, as shown in Figure 12e. In addition, in Figure 12a, many white particles were uniformly dispersed in the matrix within the red outline. The continuously distributed coarse carbides at grain boundaries were $M_{23}C_6$ carbides, which was due to the decomposition of MC to $M_{23}C_6$ after solid solution treatment (MC + γ → $M_{23}C_6$ + γ′) and the precipitation of secondary $M_{23}C_6$ carbides at eutectic fronts or grain boundaries during long-term aging treatment, as shown in Figure 12f. Previous studies have shown that carbides are accumulated at grain and subgrain boundaries, which reduces the grain boundary plasticity and increases the risk of cracks generation.

*3.4. Mechanical Properties*

3.4.1. Nanoindentation

The load-displacement curves of the K438 alloy with different scanning strategies and post-treatment conditions are depicted in Figure 13. It can be observed from the graphs that the plastic phase curves for S1 and S2 had almost equal lengths and slopes at maximum load, indicating that the parts under these two scanning strategies had similar plastic behavior. However, the plastic phase curve of HTS1 post-treatment was not much different from that of S1, indicating that post-treatment had little effect on the plasticity of S1. In contrast, the plastic stage curve of HTS2 became steeper, indicating that the post-treatment reduced the plasticity of S2. After post-treatment with different scanning strategies, a large number of square γ phases disappeared, while fine secondary γ phases precipitated and grew uniformly in the matrix, resulting in an increase in the Young's modulus of the alloy. Five imprints were taken for each sample, and the data in Table 2 were obtained by averaging the five values. The Young's modulus of HT S1 and HT S2 increased by 18 GPa and 27 GPa, respectively.

The island scanning strategy employed a square scanning area and a 90° rotation for each layer, which impacted the hardness. However, the laser overlap region experienced more crack propagation, which made suppression difficult. In contrast, the 'back-and-forth' scanning strategy changed the laser scanning direction significantly by 67° after creating a layer, which limited crack expansion. The morphological and distribution analyses of the crack revealed its control of the changing direction of neighboring molten pool dendrites. The Young's modulus of the alloys increased with different scanning strategies, while

the hardness significantly decreased after post-treatment. The hardness of HT S2 was reduced to 8.2 GPa. In the post-treatment process, a significant number of square γ′ phases disappeared, while fine secondary γ′ phases precipitated and grew uniformly in the matrix. Accordingly, the strength of the alloys was effectively improved. However, during the heat treatment of solid materials, particularly high-temperature annealing or aging treatment, the primary grains continued to grow. This growth resulted in an increase in grain size, which led to a decrease in the alloy's hardness, as demonstrated in Figure 11.

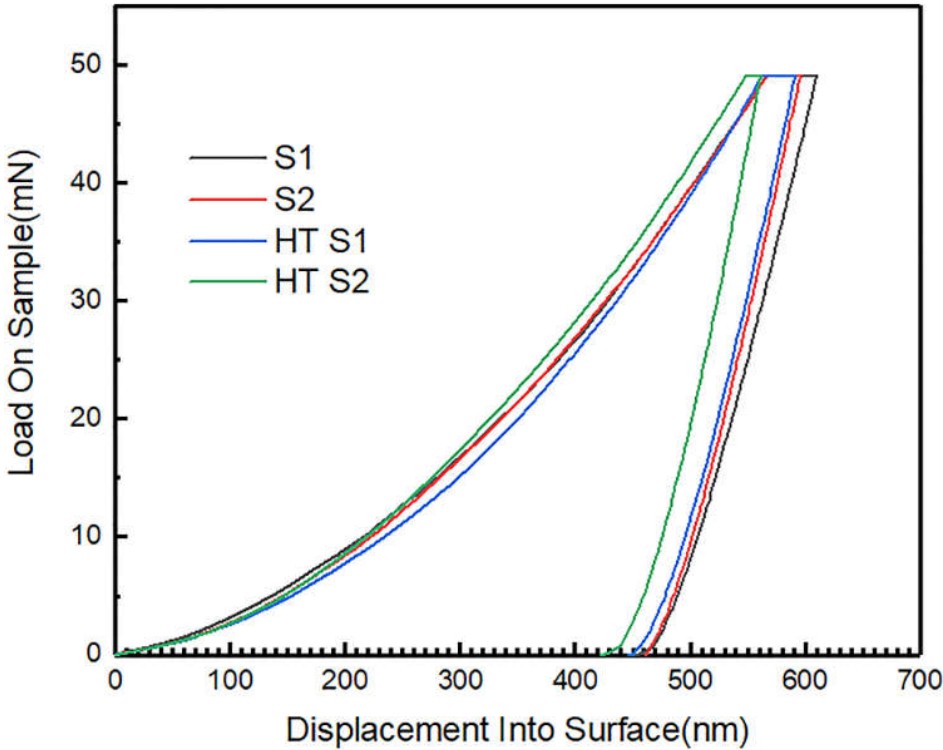

**Figure 13.** Load displacement curve under different scanning strategies and post-treatment.

**Table 2.** Hardness and Young's modulus of samples under different scanning strategies and post-treatment.

| Sample | Hardness (GPa) | Young's Modulus (GPa) |
|---|---|---|
| S1 | $9.6 \pm 0.2$ | $179 \pm 3$ |
| S2 | $8.7 \pm 0.3$ | $190 \pm 1$ |
| HT S1 | $9.1 \pm 0.2$ | $197 \pm 2$ |
| HT S2 | $8.2 \pm 0.1$ | $217 \pm 2$ |

After post-treatment, the huge square γ phase in the alloy disappeared and the fine secondary γ phase precipitated, grew, and was uniformly dispersed in the matrix, as shown in Figure 12a,b. At the same time, the K438 alloy was mainly affected by precipitation strengthening. Uniform distribution of fine carbides hinders dislocation motion, which increases the ability of precipitation strengthening and improves the strength of the SLMed K438 superalloy. After aging treatment, many uniformly recrystallized grains adhered to the original grain boundaries and precipitated out. The large amount of $M_{23}C_6$ carbides precipitated on the grain boundary increased the sensitivity of the cracks, increased the initiation of liquefaction cracking along the grain boundary, and reduced the plasticity of the alloy.

### 3.4.2. Tensile Properties

The 'back-and-forth' scanning strategy with a 67° interlayer rotation angle effectively suppressed crack initiation and propagation in the SLMed K438 alloy. The strength of the alloy was effectively maintained, resulting in a higher tensile strength (1309 MPa), as shown in Figure 14. The irregular arrangement of cellular crystals with different orientations in the island scanning strategy led to a diversity of crystal growth directions. This variety allowed the material to undergo plastic deformation in multiple directions during stretching, which enhanced the material's ductility [53,54]. Compared to the 'back-and-forth' scanning strategy, the island scanning strategy exhibited a higher ultimate strain (3.24%). This was because the island scanning strategy involved a 90° rotation direction between adjacent islands, enabling the internal stresses in the two directions within the alloy to be constrained by each other in the positive stress stretching state, which enhanced the toughness of the alloy. Nevertheless, the island scanning strategy led to an increase in cracks at the overlap, which caused the tensile strength of the alloy (1256 MPa) to decrease, as shown in Figure 14. In contrast, the tensile strength of HTS1 and HTS2 was higher than that of S1 and S2 because of the uniform distribution of MC carbides and $\gamma'$ phases in the matrix after post-treatment.

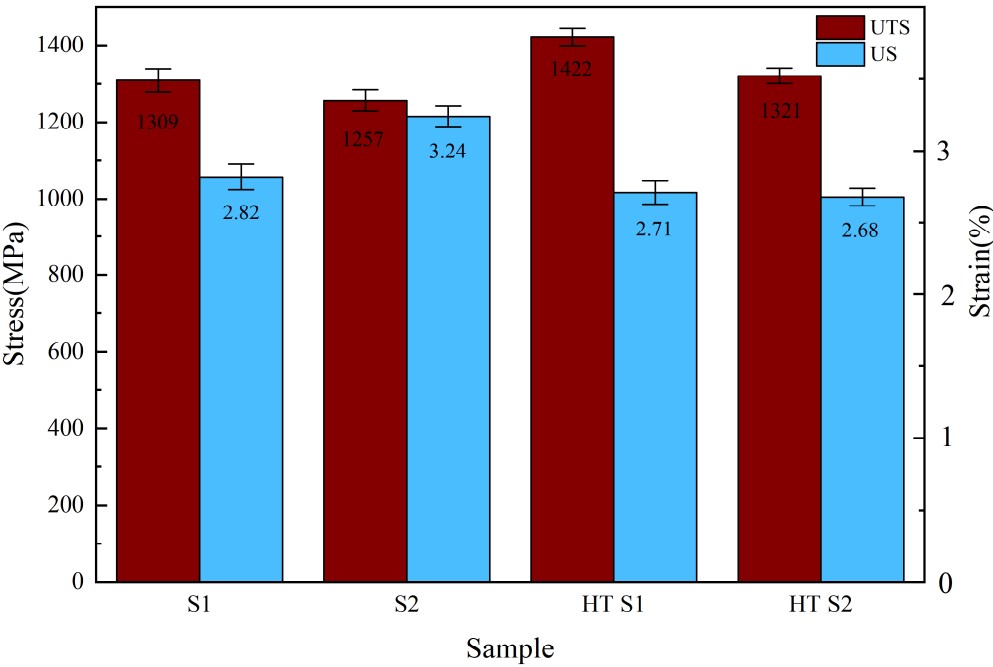

**Figure 14.** Tensile data with different scanning strategies and heat treatment.

During the solid solution treatment stage, the large $\gamma'$ phase inside the alloy melted, resulting in the transformation of several diffusely distributed MC carbides into $M_{23}C_6$ carbides. In contrast, during the aging treatment stage, the precipitation of secondary $M_{23}C_6$ carbides at the grain boundaries was accompanied by the uniform precipitation of fine $\gamma'$ phases throughout the alloy. As these precipitated phases were uniformly distributed, cracks propagated between different phases, diminishing the stress concentration region and thus enhancing the ultimate tensile strength of the alloy. Although the precipitated phase out could hinder the dislocation motion and improve the overall strength of the alloy, the stress concentration increased the sensitivity of the grain and sub-grain boundary cracking, produced more strain aging cracks, and reduced the ultimate strain of the alloy.

To carry out a thorough analysis of the tensile properties of the samples in the as-built state and after post-treatment using different scanning strategies, we examined the fracture characteristics through micro-fracture morphology, as illustrated in Figure 15. The fracture surfaces of specimens, whether in the as-built state or after post-treatment, predominantly displayed quasi-cleavage planes and cleavage ladders. These characteristics

were significantly correlated with reduced elongation. The fracture process involved two crucial stages, namely, the formation and extension of cracks [55].

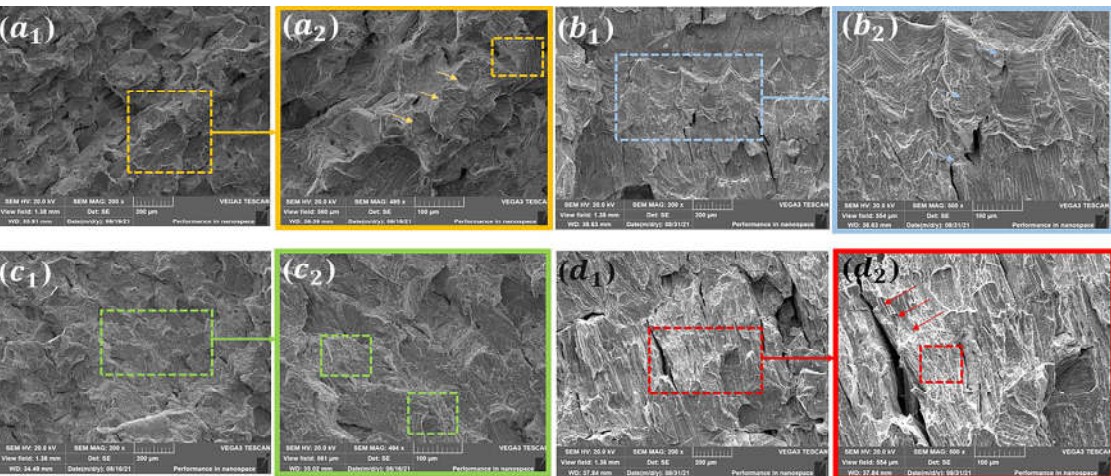

**Figure 15.** SEM images of tensile fracture morphology. (**a1**,**a2**) S1 sample, (**b1**,**b2**) S2 sample, (**c1**,**c2**) HTS1 sample, (**d1**,**d2**) HTS2 sample.

On the fracture surface of S1 shown in Figure 15(a2), there were a few dimples. These dimples were generated while deforming the specimen and grew rapidly along the tearing prongs, eventually forming a perforated fracture indicating ductile fracture mode. However, quasi-cleavage planes and cleavage ladders were observed in the region pointed out by the arrows. This implies that the fracture surfaces separated along specific crystalline surfaces, indicating the presence of brittle fracture modes. The 'back-and-forth' scanning method resulted in a combination of ductile and brittle fracture modes. The fracture surface of S2 shown in Figure 15(b2) reveals numerous small, irregular cracks expanding along the grain boundaries in the direction of the main cracks, resulting in herringbone shapes and cleavage ladders, which is a characteristic of the brittle fracture mode. The brittle fracture was related not only to the distribution of the second-phase particles but also to the high internal stress at the laser overlap caused by the island scanning strategy, which promoted crack extension and the formation of brittle fracture patterns [56]. Despite the presence of dimples in some areas, brittle fracture dominated, causing the island scanning strategy to display a mixed fracture pattern.

In summary, there is a significant relationship between the fracture mode and the scanning strategy. This could be attributed to the high-speed scanning and cooling rate of the laser, which resulted in the uneven microstructure of the material. Distinct variations in grain size and grain boundary distribution may have existed in different regions, causing different ductile and brittle properties in local areas, which, in turn, resulted in a mixed mode of ductile and brittle fracture. As a result, the SLM process frequently exhibited mixed fracture modes in selective laser melting [57,58].

In Figure 15(c2), the area of the dimples in the cracks after post-treatment is significantly reduced, and the brittle fracture part is increased, indicating that the precipitation of secondary $M_{23}C_6$ carbides at the grain boundary and the stress concentration reduced the plasticity of the alloy, which is consistent with the previous ultimate strain data. As shown in Figure 15(d2), due to the existence of a large number of cleavage ladders, there was a typical transgranular fracture feature, indicating that the plasticity was greatly reduced. The generation and extension of the cracks were mainly concentrated in the interior of the grains, which hindered the maintenance of the elongation of the sample.

## 4. Conclusions

(1) Under the 'back-and-forth' scanning strategy, the material achieved the highest density and hardness of 99.74% and 9.61 GPa, respectively. In contrast, the island scanning strategy resulted in a density of 99.34% and hardness of 8.70 GPa. The interlayer rotation of the 'back-and-forth' scanning strategy made the molten pool more uniform and dense, thereby improving the density and hardness.

(2) Liquefaction cracking (LC), solidification cracking (SC), and ductility dip cracking (DDC) were produced during printing. In the two scanning strategies, the interlayer rotation angle of 67° and the partition scanning characteristics helped to reduce the generation of cracks. On the contrary, the longer scanning line length and the overlapping area of the laser easily caused cracks. The scanning vector directly affected the crack vector.

(3) The interlayer rotation of the 'back-and-forth' scanning strategy formed a shorter dendritic structure. In contrast, the partition characteristics and diversified heat dissipation directions of the island scanning strategy led to the anisotropy of the grains, and the plasticity reached the highest at 3.24%.

(4) After post-processing, stress was easily released at the crack tip and produced strain-age cracking. The MC phase was decomposed into $M_{23}C_6$ and precipitated at the grain boundary, which reduced the plasticity of the material. The misorientation of grain was reduced, and the secondary $\gamma'$ phase precipitated and grew to be uniformly dispersed in the matrix. The highest Young's modulus (217.44 GPa) and the highest tensile strength (1421.71 MPa) of the material were achieved.

**Author Contributions:** B.Z., Z.X., H.Y. and H.S. conceived the idea; Z.X. and B.Z. conducted the experiments and analyses; B.Z., H.Y., P.Z., Q.L. and H.S. provided the data; all authors contributed to the writing and revisions. All authors have read and agreed to the published version of the manuscript.

**Funding:** This research was funded by China Postdoctoral Science Foundation (2021M7010380) and Class III Peak Discipline of Shanghai—Materials Science and Engineering (High-Energy Beam Intelligent Processing and Green Manufacturing).

**Institutional Review Board Statement:** Not applicable.

**Informed Consent Statement:** Not applicable.

**Data Availability Statement:** Data is contained within the article.

**Conflicts of Interest:** The authors declare no conflict of interest.

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
