# Peer review of "Influence of Scanning Strategy and Post-Treatment on Cracks and Mechanical Properties of Selective-Laser-Melted K438 Superalloy"

_coatings, doi:10.3390/coatings14040414_

Round 1

Reviewer 1 Report

Comments and Suggestions for Authors

Dear Authors,

This manuscript presents an exploration of the intriguing field of 3D metal printing. The overall content demonstrates potential for publication. However, we recommend addressing a few minor aspects to further strengthen the work:

- Figure 2: Enlarging Figure 2 would enhance clarity and allow readers to better visualize the presented information.

- 3D Printing Machine Details: A more comprehensive description of the utilized 3D printing machine would be beneficial for a thorough understanding of the research context.

- Figure 3: Including both a real photograph and a 3D view of the tensile testing sample would significantly improve the informative value of Figure 3.

- Scanning Strategy:

The application of two distinct scanning strategies within the research holds particular interest. We kindly suggest elaborating further on these two options (lines 168-175) to provide a clearer understanding for the audience.  Additionally, a more in-depth review of existing scanning strategies within the introduction would provide valuable context for the presented research.

Sincerely,

Reviewer 2 Report

Comments and Suggestions for Authors

When preparing the manuscript about the effect the scanning strategy and post-treatment on crack formation, microstructure and mechanical properties of selective-laser-melted K438 superalloy, the authors did not always pay attention to the terms used, the selection of appropriate vocabulary and lose logic when discussing the results. Despite the large amount of painstaking experimental research, it remains unclear what the aim of the work was, why it was undertaken. To improve the manuscript so that it has a chance to be published, the authors should pay attention to the following points:

 (3) Please remove (SLMed)

(130…132) The authors mention the optimization of SLS and post treatment parameters to achieve superior microstructure and mechanical properties. It should be clarified which parameters of the microstructure and which values of mechanical properties the authors consider superior in order to strive to achieve them? What the aim of the work was?

(141…143) Instead of a false statement, the authors should describe the defects in the morphology of spherical particles shown in Fig.1

 (158-159) Insert missing words [solid] solution treatment, [heat] treatment, for common terms. Apply them for the whole manuscript

(161…163) (176…177) (187…189) Unclear sentences, re-formulate them!

(167) Immutable parameters (Laser parameters & post treatment columns) should be removed from Table 2 and moved to its title or text

(205)…the phase composition of the coating, using a Cu target ….. Why are coatings and targets mentioned here?

(208) Checkup, what kind of indenter was used?

Fig. 12 e d  Convert a negative image into a positive one.

(570) Specify how many imprints were made to obtain the data in Table 3. Also specify the mean square deviation for the values of hardness and modulus of elasticity. Note that it makes no sense to give more than one sign after the decimal separator for hardness, and use only integers for the modulus of elasticity

Comments on the Quality of English Language

The authors did not always pay attention to the terms used, the selection of appropriate vocabulary. To improve the manuscript  the authors should pay attention to the following points:

 (158-159) Insert missing words [solid] solution treatment, [heat] treatment, for common terms. Apply them for the whole manuscript.

(161…163) (176…177) (187…189) Unclear sentences, re-formulate them!
